# Dual Detection of the Chytrid Fungi *Batrachochytrium* spp. with an Enhanced Environmental DNA Approach

**DOI:** 10.3390/jof7040258

**Published:** 2021-03-30

**Authors:** David Lastra González, Vojtech Baláž, Jiří Vojar, Petr Chajma

**Affiliations:** 1Department of Ecology, Faculty of Environmental Sciences, Czech University of Life Sciences Prague, Kamýcká 129, 65 00 Prague-Suchdol, Czech Republic; vojar@fzp.czu.cz (J.V.); chajmap@fzp.czu.cz (P.C.); 2Department of Ecology and Diseases of Zoo Animals, Game, Fish and Bees, Faculty of Veterinary Hygiene and Ecology, University of Veterinary and Pharmaceutical Sciences Brno, Palackého tř. 1946/1, 612 42 Brno, Czech Republic; balazv@vfu.cz

**Keywords:** *Bd*, biomonitoring, *Bsal*, chytridiomycosis, fungal pathogens, water samples

## Abstract

Environmental DNA (eDNA) is becoming an indispensable tool in biodiversity monitoring, including the monitoring of invasive species and pathogens. Aquatic chytrid fungi *Batrachochytrium dendrobatidis* (*Bd*) and *B. salamandrivorans* (*Bsal*) are major threats to amphibians. However, the use of eDNA for detecting these pathogens has not yet become widespread, due to technological and economic obstacles. Using the enhanced eDNA approach (a simple and cheap sampling protocol) and the universally accepted qPCR assay, we confirmed the presence of *Bsal* and *Bd* in previously identified sites in Spain, including four sites that were new for *Bsal*. The new approach was successfully tested in laboratory conditions using manufactured gene fragments (gBlocks) of the targeted DNA sequence. A comparison of storage methods showed that samples kept in ethanol had the best DNA yield. Our results showed that the number of DNA copies in the Internal Transcribed Spacer region was 120 copies per *Bsal* cell. Eradication of emerging diseases requires quick and cost-effective solutions. We therefore performed cost-efficiency analyses of standard animal swabbing, a previous eDNA approach, and our own approach. The procedure presented here was evaluated as the most cost-efficient. Our findings will help to disseminate information about efforts to prevent the spread of chytrid fungi.

## 1. Introduction

Species detection is an integral—but often laborious—task in any biological field research. Environmental conditions and species-specific ecology (e.g., hidden way of life, scattered distribution, or small size) can substantially decrease the probability of certain taxa being detected [1,2]. Technological progress in the last 15 years has enabled the collection and analysis of DNA traces left by organisms in the environment (eDNA) and has led to a revolution in biodiversity studies. Species can be detected from soil, sediment, or water without any direct signs of their presence [3]. The method is broadly used for multiple purposes, for detecting elusive [4], endemic [5], or invasive species [6]. At the same time, multiple infectious diseases with a detrimental impact on biodiversity have emerged and have been recognized on a global scale [7,8,9,10]. Amphibians have become the canaries in the coal mine in this pathogen-mediated crisis [11], with chytridiomycosis being the main culprit in their decline [12]. This disease is caused by two chytrid fungal species: *Batrachochytrium dendrobatidis* (*Bd*) [13] and *B. salamandrivorans* (*Bsal*) [14].

Amphibian chytrid fungi are potentially good targets for the application of eDNA detection, owing to their microscopic size, the large quantities of actively dispersing zoospores that are usually produced, occupation of an aquatic environment, and the ready availability of established detection assays. Attempts to use eDNA detection in the case of *Bd* [15,16,17,18] or *Bsal* [19] are still surprisingly sparse, although qPCR [20,21], nested PCR [22], and standard PCR [14] assays are sensitive enough to detect the DNA equivalents of individual cells. The reasons limiting the broader use of eDNA in the detection of both chytrid fungi include the perceived difficulty of the procedure, inconsistent results in comparison with standard individual sampling protocols [15], and the potentially high cost of sample collection and processing [19].

The crucial factor in eDNA detection of target pathogens is the establishment of an applicable procedure for eDNA collection, sample storage and DNA isolation. Critical considerations to ensure quality assurance and minimum requirements stated in [23] should be followed, and the current lack of unified criteria for procedures makes comparison of results challenging. As an example, various types of filters are used—polycarbonate track-etched [17], polyethersulfone [18], nitrate cellulose [15], or VigiDNA^®^ (Spygen, Le Bourget du Lac Cedex, France), unspecified material [19]. At the same time, the current rapid spread of the pathogens requires simple, fast, robust, and cost-effective surveillance methods. In order to meet these requirements, particular aspects of sample collection and subsequent DNA extraction should be considered: the pore size of the filter, the type of pumping system, the extraction kit and complementary kits (e.g., an anti-inhibitor kit or a purification kit), and the storage method. An extensive review of the range of the above-mentioned parameters within 36 articles can be checked in Appendix A.

The choice of the storage method requires special attention. The stability of the eDNA in the filter after collection has a strong effect on species detection [24,25], and multiple methods of sample preservation have already been tested. The ideal sample conservation method should be technically simple and applicable in field conditions, should provide stability of the sample in ambient temperatures, should be safe to use and transport, and should be cost-effective. The most widely used storage methods are freezing and 96% ethanol (see Appendix A). However, freezing the samples reduces the yield [25], and ethanol evaporates easily even when the enclosed filters are sealed with a cap. Moreover, it is necessary to use a pipette and tips in field conditions, or to transport a portable freezer. Some articles have reported studies on this crucial topic [24,25], but they were not linked to the detection of *Batrachochytrium* spp. and did not include any other affordable storage method, such as silica gel.

Lastly, the economic aspect of any method greatly affects its application potential and how widely it is used. Unfortunately, only a small number of publications have included precise costs or estimates of the costs of their working procedure that would enable non-affordable materials to be rejected, either due to their unavailability or because their high cost makes them unusable when large amounts are needed (but see [4,23,26]).

The purpose of this research is therefore to find an accessible protocol for detecting *Bsal* with an eDNA approach. Subsequently, the aim is to make a protocol that is compatible for detecting *Bd* at the same time, thus creating the first protocol, to our knowledge for detecting both chytrid fungi by eDNA. A further goal was to identify the best storage method for the filters that will be used. Finally, we have calculated a strict comparison (materials, costs, protocol complexity) of existing protocols to optimize the resources and thus to strengthen efforts to detect these pathogens and promote further research.

## 2. Materials and Methods

### 2.1. Study Area and Collection of Field Samples

Since 2017, we have been monitoring by swabbing several caudate populations in the northern part of Spain, which have been positive for *Bsal* [27] or see Table 1. To collect samples for the purposes of this research, we swabbed amphibians and filtered water from the habitat where the animals were collected. We tested the filters in the *Bsal*-positive localities mentioned in [27], and also in four other localities. We also sampled several Czech localities, near Staré Město, Zlín Region and Sokolov, Karlovy Vary region, where we collected eDNA samples for an evaluation of the presence of *Bd*. In total, 47 filters and 148 swabs were collected in 11 localities (see details in Table 1).

Skin swabs were collected following the standard procedure for sampling amphibian chytrid fungi, as described in [28]. The eDNA was collected using SVHVL10RC filters, 0.45 µm pore size, PVDF membrane, with a Luer outlet (Millipore Sigma, Burlington, MA, USA). These filters were attached to a 50 mL Omnifix Luer Lock syringe (B. Braun, Melsungen, Germany). Each syringe was first rinsed with water from the tested waterbody, was then filled completely with water pre-filtered by nylon fabric (to avoid macroscopic organic material), was attached to the filter and was manually forced to pass through. Pre-filtration was done just if organic material (e.g., leaves, algae, and sticks) could block the inlet or the outlet of the filter device (not affecting the capture of zoospores). A specific example of this situation was when a layer of *Lemna minor* covered the water surface. Multiple repetitions of 50 mL with water from all around the perimeter of the water body were carried out, until the filter was clogged. Then the remaining water in the filter was removed entirely by pushing air with the syringe. The maximum number of repetitions was 20, giving 1000 mL of water filtered per water body. The collected filter samples were preserved with Longmire’s buffer and silica gel (see extended information in Appendix A). A hygienic protocol was used, including a different set of gear (e.g., gloves, boots, or nets) for each locality. The set of gear was disinfected when the procedure had been completed.

### 2.2. DNA Extraction and qPCR

The filter membrane was removed from the filter case and was cut into smaller fragments to fit in a 2 mL tube. We extracted the DNA with the ISOLATE II Genomic DNA Kit (Bioline, Meridian Bioscience Inc., Cincinnati, OH, USA), with the following modifications of the protocol: during the pre-lysis step, the lysis step and the step for adjusting the DNA binding conditions, we triplicated the volume of each reagent to cover the filter completely. The DNA binding step with 96% ethanol was performed in a clean 2 mL tube with all the liquid collected from the previous lysis step. The elution step was performed in two consecutive rounds of 50 µL to increase the yield. The isolated DNA was stored in 1.5 mL tubes at −20 °C.

The detection of *Bd* and *Bsal* DNA followed the protocol of [21]. To avoid false negative results due to PCR inhibition, we ran each eDNA sample with TaqMan™ Exogenous Internal Positive Control (IPC) Reagents (Thermo Fisher Scientific, Waltham, MA, USA). This could not be performed in the same reaction as the detection itself, as the IPC and *Bsal* probe share the same fluorophore parameters. All analyses were performed in duplicate in Roche LightCycler 480 II (Roche Diagnostics, Prague, Czech Republic), using the Roche Probes Master mix (Roche Diagnostics, Prague, Czech Republic). Only in the case of the Czech samples, we used a simplified qPCR assay targeting only *Bd* (using the cycling program after [21], but the mix contained only *Bd* primers and a probe). As the quantification standards for *Bd*, we used 100, 10, 1, and 0.1 GE per 5 µL (standard volume added in the PCR reaction) dilutions of *B. dendrobatidis* genomic DNA, from the GPL lineage, strain IA042, provided to us by Trenton Garner, from the Institute of Zoology, Zoological Society of London (United Kingdom). Quantification standards of *B. salamandrivorans* of 100, 10, 1, and 0.1 GE per 5 µL were made from DNA provided by An Martel, of Ghent University (Belgium). Later, we adopted gBlocks (see gBlock in vitro testing) as *Bsal* standards. In addition, some samples were sent to Trier University in order to confirm our results. A subset of samples from field and laboratory experiments were retested with a recently-developed assay for *Bsal* eDNA testing [19].

In all qPCR tests, the sample was considered positive only if both wells amplified, the increase in fluorescence showed a standard sigmoidal curve, and the Ct value was below 40. Samples with single well amplification were retested.

### 2.3. gBlock In Vitro Testing

We used Genbank sequences of *Bsal* (KC762295.1 and NR_111867.1) in designing gBlock (Integrated DNA Technologies Inc., Coralville, IA, USA) fragments to be used as a substitute for free eDNA in spiked water tests. We used a *Bsal* gBlock 213 bp in length that overlaps with the *Bsal* primers [21] target sequence by 26 bases on both sides. The same gBlock is also applicable with recently published eDNA qPCR [19]. Positions 96 and 97 contain two adenine nucleotides in the position of thymine in the reference sequences, in order to identify possible contamination by the gBlock in the event of equivocal results suspected to have been caused by laboratory cross-contamination.

To test the filtering and the subsequent qPCR sensitivity in detecting *Bsal* eDNA, we made five 500 mL samples of natural pond water spiked with 10^2^, 10^3^, 10^4^, 10^5^, and 10^6^ gBlock copies. The water samples were filtered and the eDNA was isolated on the day of spiking. *Bsal* gBlock sequence: 5′ CAGAACTCAGTGAATCATCGAATCTTTGAACGCACATTGCACTCTACTTTGTAGAGTATGCCTGTTTGAGAATCAATAGTATTTTCTTGTTCTATaaTCTTTTTTTAATTCATTTCCTTGTCTTTTTATATCATCTAAAAAGTGATATAAAAATAGGGTTAGGGATGAAGAGGGGGAGATGGAGCAGATAATGAGTGATTAGTTGAGGTTCT 3′.

The genomic equivalents (GE) were calculated with the use of gBlocks with 10-fold serial dilutions from 10^0^ to 10^8^ copies per 1 µL. We ran these standards in quadruple repetitions in separate qPCR [21], with 5 µL of the sample and 20 µL of the mastermix in reaction. The produced standard curve was saved and was used to quantify the *Bsal* DNA load of the sample and a single standard concentration was used as the calibrator.

In the case of performing the assay developed by [19], we used a new set of gBlock standards run in triplicate with the samples and used the in-run standard curve for quantification. In all analyses, we used the calculation of the 2nd derivative maximum, which is available in LightCycler 480 software.

To make our quantification results easier to compare with widely used genomic DNA standards, we compared the gBlock standard curves with the *Bsal* genomic DNA standards, and we calculated how the gBlock copy numbers translate to *Bsal* GE values. For this purpose, we used data available from the *Bsal* detection laboratory ring test performed in 2017 in collaboration with *Bsal* reference laboratory at Ghent University. The equation for the standard curve generated on the basis of gBlocks (5 × 10^0^, 5 × 10^1^, 5 × 10^2^, 5 × 10^3^, 5 × 10^4^, 5 × 10^5^, 5 × 10^6^, 5 × 10^7^, and 5 × 10^8^) was used to calculate the copy number of ITS based on the Ct value of each genomic *Bsal* standard (10^−1^, 10^0^, 10^1^, and 10^2^ GE per reaction). For each genomic standard, the resulting Ct value was translated into copy numbers, corrected to the given dilution and averaged.

The calculation of the efficiency of the eDNA capture was based on the detected copy number, multiplied by 20 to account for the dilution factor during the DNA extraction (total elution into 100 µL, 5 µL of undiluted sample used in reaction), and then divided by the original number of copies in the water.

### 2.4. Storage Experiment for eDNA Samples

We compared widely-used filter storage methods, i.e., Longmire’s buffer (LB), silica gel (S), ethanol (EtOH), together with no storage method as a control (kept at room temperature). A preliminary test of the filter storage method was performed using five liters of water from a pond with amphibian presence but no chytrid fungi, collected in November 2019. The water was pre-filtered to remove the abundant plankton (e.g., *Daphnia* sp., *Ephemeroptera* larvae) and larger debris. We spiked four bottles containing 0.5 L of pond water with 106 copies of *Bsal* gBlock each. Filtration was performed as explained above (see Study area and collection of field samples). One filter was preserved by each of the compared storage methods (LB/S/EtOH/control). To ensure repeatability of the methodology, note the following specifications: LB and EtOH were added in an amount of 2 mL pipetted inside the filter; S was 2–5 mm with an orange indicator (P-lab, Prague, Czech Republic); 96% EtOH and control (no storage method applied). LB and EtOH were stored with Luer-Lock caps, and all of them were placed inside a Falcon tube. The filters were stored at room temperature for 9 weeks. DNA extraction and qPCR analyses were carried out as stated above (see DNA Extraction and qPCR).

The second test of storage methods used more repetitions and was slightly modified. We used 400 mL of water from the same pond collected in April 2020, and we again spiked it with 106 copies of *Bsal* gBlocks. Each storage method was used on three filters. In filters stored in S, we cracked and removed the external layer of the filter and we left the filter in direct contact with the silica gel. The LB and EtOH filters were sealed with Luer Lock caps and all of them were placed inside a Falcon tube. The filters were stored at room temperature for 6 weeks. DNA extraction and qPCR analyses were carried out as stated above (see DNA Extraction and qPCR).

### 2.5. Cost Assessment

In order to assess the cost-efficiency of traditional sampling (swabbing), together with our approach and a previously used approach for detecting *Bsal* with eDNA [19], we performed a cost comparison among these sampling protocols. All prices (in euros) and all costs were calculated in July 2020, with the exception of the cost of the services provided by the SPYGEN laboratories in April 2020. If the price was in another currency, the exchange rate valid on 13 July 2020 was used. The budget was envisaged for a single locality, using one filter or swabbing 20 animals, and for two people. The laboratory extraction costs are for one replicate. This calculation covered perishable materials and reagents, but labour costs, transport, usage of university facilities and long-lasting materials (e.g., traps, buckets, and disinfection equipment) were not included. We considered these conditions to be the minimum required for developing a complete analysis with total guarantees.

### 2.6. Statistical Analyses

Filter storage methods (fixed effect) and their effect on the retention of DNA expressed by the Ct values (response) were compared using linear mixed effects models with a Gaussian error structure and a filter as a random intercept [29]. Model-based Bayesian 95% credible intervals of Ct were obtained using the 2.5 and 97.5 percentiles from the posterior distribution of 5000 simulated values [30]. For the purposes of the analyses, Ct values of 0 and 40 or higher were treated as negative and were all set to 40. All statistical analyses were performed in R, version 3.5.3 [31].

## 3. Results

### 3.1. Field Sampling

Out of 13 filters collected in Spain, 12 were positive for *Bsal*. These filters were from nine different localities, in four of which this was the first case of *Bsal* detection (see Table 1). Moreover, in eight localities we were able to test both sampling options: filters and swabs. Just in three of the localities, both eDNA and swabs tested positive for *Bsal*. This means that eDNA was able to prove the presence of *Bsal* in five localities where no positive trace had been found with the swab methodology.

The analysis of the eDNA detected the presence of both amphibian chytrid fungi in three filters in Spain, two from the same locality (see Table 1). This is to our knowledge the first time that both *Batrachochytrium* species have been detected by eDNA within a single sampled site. In addition, one filter in the Czech Republic tested positive for *Bd* (see Table 1). All IPC tests showed no difference among the tested samples and produced typical amplification curves.

### 3.2. gBlock In Vitro Testing

The comparison between the gBlock standards (standard curve y = −1.478ln(x) + 40.221, R^2^ = 0.998) and genomic DNA resulted in a mean value of 121.3 copies per cell (SD = 33.1). The targeted ITS region therefore seems to be present in the *Bsal* lineage used for producing genomic standards in approximately 120 copies.

The efficiency of eDNA collection evaluated by qPCR testing of pond water filters spiked with 10× serial dilutions of *Bsal* gBlocks (10^2^–10^6^ copies per 500 mL filtered water) showed that amplification occurred consistently only in 10^4^ and higher numbers of gBlock copies. One well of 10^3^ copies was positive with Ct 39, while the 10^2^ copies per filter were not detectable. The overall gBlock capture (proportion of detected copies by copies added to the spiked water prior filtration) from spiked pond water using freshly isolated filters that tested positive was 20%. However, the lowest concentration equivalent of approximately a single cell did not amplify at all.

### 3.3. Storage Experiment for eDNA Samples

In the preliminary test, we detected *Bsal* gBlocks after nine weeks in all filters. The filter that was preserved in ethanol got the earliest response. In the subsequent test with three replicates, the ethanol filters were again the first to amplify, followed by the filters preserved in silica gel, then LB, and finally the control filters (see Appendix A). The same results appeared with the primers from [19]. The filter storage methods had a significant effect on the retention of DNA expressed by the Ct values (primers from [21]: F = 7.59, *p* = 0.01, R^2^_LMM(m)_ = 0.70, R^2^_LMM(c)_ = 0.94; primers from [19]: F = 8.40; *p* = 0.007, R^2^_LMM(m)_ = 0.43, R^2^_LMM(c)_ = 0.83). While point estimates from both models showed that the filters stored in ethanol were first to amplify, followed by the filters preserved in silica gel, Longmire’s buffer and the control, Bayesian 95% credible intervals showed that only ethanol and silica gel were substantially better storage methods than the control (Figure 1).

### 3.4. Cost Assessment

A cost comparison of the sampling protocols between traditional sampling (swabbing), a previously used approach for detecting *Bsal* with eDNA [19], and our approach presented here revealed our methodology as the most economical option (see Appendix A), with an approximate cost of 22 euros per locality. In comparison, the costs for the protocol developed in [19] are approximately 72 euros per sample (locality). However, the analysis performed by a private laboratory had a dramatic influence on these costs (see the footnote in Appendix A). Note that the VigiDNA^TM^ filter cost five times more than our equivalent gear. On the other hand, we added the costs of traditional swabbing monitoring, either the protocol with the Qiagen Blood and Tissue kit (Qiagen, Hilden, Germany) or PrepMan^TM^ (Thermo Fisher Scientific, Carlsbad, CA, USA). In these cases, in order to make a fair comparison, we calculated the costs for 20 samples and gear for two people. With these conditions, the cost per locality for the Qiagen Blood and Tissue kit is around 81 euros, and when PrepMan^TM^ is used the cost is approximately 27 euros per locality (see Appendix A).

## 4. Discussion

Our results have confirmed *Bsal* in previous positive localities from [27] and we have announced four new positive sites in Spain. Our proposed eDNA collection procedure also detects *Bd*. We have calculated the number of ITS1 rRNA gene copies present in one cell, i.e., 120, and showed that 96% ethanol was the best storage method.

We again found the five *Bsal* positive sites from [27], and we have detected four new ones. We observed inconsistency in the detection through the years, as was also found in [32,33]. In Germany, due to the current reduced numbers of adult fire salamanders (*Salamandra salamandra*) it is becoming harder to obtain samples to test the pathogen [33]. In addition, almost entire populations can be wiped out from an area after a severe *Bsal* outbreak [34], and swabbing adult individuals is therefore becoming quite a challenging task. It was imperative to develop a protocol that is not dependent on locating amphibians and that unifies the latest advances in laboratory techniques for detecting *Bsal*. However, it is important to note that eDNA detection is not able to discriminate between the presence of viable cells/organisms and the presence of residual DNA fragments [26]. It is not advisable to collect DNA after heavy rains or after any event that disturbs streams or water bodies [35]. Sediments could re-suspend eDNA up to six months later, and this could give a false positive for the presence of species [36]. A comparison between confirming the presence of organisms by eDNA and by traditional monitoring showed that, for example, four visits for amphibian monitoring in Mediterranean ponds were needed to obtain similar detectability as eDNA [37]. Nonetheless, in order to obtain 95% probability of detecting *Bd* by eDNA in a site with confirmed positive amphibians, it was necessary to get four samples of 600 mL or five samples of 60 mL [38].

Since the first chytridiomycosis outbreaks were reported, changes have been made in the methodology for swab sample processing before detecting amphibian chytrid fungi. *Bsal* DNA was originally isolated from swabs with the use of PrepMan^TM^ (Thermo Fisher Scientific, Carlsbad, CA, USA) (recommended by [21,39]). However, spin-column based DNA extraction kits seem to outperform the simple and affordable option of PrepMan^TM^ [40,41], because sites with amphibians hosting *Bd* or *Bsal* in low intensities may be falsely identified as negative if the less efficient DNA extraction method is used [41,42].

Despite several recommendations and methodological publications on *Bd* and *Bsal* detection [28,39,43], there are ongoing inconsistencies among research teams in the definition of a positive result. For example, there is inconsistency in the Ct value set as the threshold to distinguish positives from late amplification false positives (e.g., [44] uses 40 cycles, while [45] uses 45 cycles). Our data based on the known number of gBlock fragments in reaction show that a single DNA molecule with the target sequence should amplify at 40.2 cycles. We therefore considered samples as positive only if the Ct values were less than 40, and if both wells amplified. Consistency of amplification in all wells where a sample was added is not universally used (see the discussion in [23]), with many studies considering single well amplification sufficient [18,38,41,45]. Multiple repetitions of qPCR tests of a single sample as used in [19], not only waste sample material, but increase the risk of in-lab contamination and false positive results. If we had adopted the strategy of one single PCR well to claim a sample as positive, we could have reported the first *Bsal* positive in the Czech Republic. However, we believe that a single amplification should not be considered sufficient for such a serious claim. We advise that this crucial issue should be solved by using sequence identifiable gBlocks or other synthetic DNA standards, not genomic DNA of the tested pathogen. DNA identity in positive results from unexpected scenarios should be confirmed subsequently by sequencing of the amplicon. To reduce the risk of false negative results, Exogenous Internal Positive Controls should be used whenever possible, as already recommended in the past [28,39].

A qPCR assay alone provides less robust support for the presence of disease in tested animals than is provided by histopathology. However, qPCR is usually a much more sensitive method than histopathology, it can be designed to be target-specific, and together with eDNA it is applicable without directly handling the amphibians. Histopathology is not compatible with eDNA filters for confirming samples, and it cannot be used in the absence of animals. If qPCR results are consistent and repeatable, there is no apparent reason for omitting some data completely, as in [46] or in the reference website BsalEurope, or for marking them “doubtful” or “unconfirmed”, as was the case of [33] in regard to some studies [27]. Nevertheless, in order to show the consistency of our results a subset of samples also tested positive in the Trier University (see Appendix A).

Moreover, other features were considered to strengthen this approach. The use of enclosed capsule filters reduces the risk of contamination during transport and storage [47]. Single-use syringes are easier to carry than various pump systems, and they avoid the risk of cross-contamination with the tubing. We recommend 50 mL syringes with a Luer-Lock tip, as they are easy to use in difficult field conditions. The overall amount of equipment needing disinfection, and the amount of waste that is generated from single-use plastics are considerably smaller than when traditional swabbing sampling schemes are used.

The collection efficiency of eDNA observed in our lab experiment may be due to limitations of the selected gBlock approach (see Table 2). The short length of the fragments designed in our case to act as a standard in qPCR likely causes faster degradation in pond water than in water spiked with viable zoospores. Moreover, we followed the DNA isolation protocol with standard lysis times, providing excess time for degradation to take place. It is also important to note that, based on [16], the recovery of zoospores varied according to where they were diluted. The yield was higher when deionized water was used, instead of the pond water that we used. The biological activity of natural water changes in time (e.g., presence of bacterial enzymes), and this could be the reason why our two different storage experiments produced slightly different Ct values, even with the same storage options. However, one cell contains approximately 120 ITS copies, and this provides a good chance of detection even if a large amount of DNA is degraded. We believe that the collection efficiency would be higher if cells were used, but the gBlock experiment was still effective in testing the whole process. We observed slightly better results when using the assay of [19] than when using the assay of [21] in the storage tests. The explanation could again be DNA degradation, considering that the first assay targets a shorter fragment within the target sequence. As DNA is degraded predominantly from the ends, it is more likely to not get degraded over the position targeted by the primers and by the probe. We experienced that the primers used in [19] were in general more sensitive, and we therefore recommend them. Issues with the specificity of the primer should be more critically evaluated if these new primers become as widely used as the primers in [20] in the case of *Bd*.

Regarding storage methods, we found that filters preserved in ethanol were the first to amplify, followed by silica gel, Longmire’s buffer, and control samples stored at room temperature. However, the difference in performance between ethanol and silica gel was not statistically significant. In addition, silica gel is simpler to use, because no extra gear is needed. This makes the silica gel our recommended storage method. We advise putting the filter and the silica gel in direct contact with each other. We excluded from our comparison keeping filters at −20 °C until extraction, because the freeze-thaw cycle affects DNA detection [48], and the process is impractical in most fieldwork scenarios. Despite its good performance, Longmire’s buffer has only rarely been used in previous studies [24,25,49]. Perhaps, the use of certain chemicals, e.g., sodium dodecyl sulfate or sodium azide, and the precautions that need to be taken when working with them, is a constraint. However, according to our results, Longmire’s buffer was a low-performance storage method (see Figure 1).

In relation to extraction kits, we used the ISOLATE II Genomic DNA Kit (Bioline, Meridian Bioscience Inc., Cincinnati, OH, USA). The most widely-used extraction kit is the Qiagen Blood and Tissue Kit (Qiagen, Hilden, Germany) (see Appendix A), but we found the protocol of the ISOLATE II Genomic DNA Kit (Bioline, Meridian Bioscience Inc., Cincinnati, OH, USA) slightly easier. There is a difference in price with the extraction kit that we used (US$ 199 for ISOLATE II Genomic DNA Kit (Bioline, Meridian Bioscience Inc., Cincinnati, OH, USA) versus US$ 168 for the Qiagen Blood and Tissue Kit (Qiagen, Hilden, Germany), 50 preparation kits). Other more affordable options for silica spin-column DNA extraction kits are available, e.g., the Gen-aid Genomic DNA mini-kit, US$ 168; the EZNA tissue DNA kit, US$ 87. Traditional phenol-chloroform-isoamyl extraction is much cheaper. According to [24], the cost is around US$ 0.2 per sample, but additional lab equipment is required to make the process safe. In addition, [50] researched about extraction kits and detected different performance, showing that it is another key parameter for the detection of chytrids.

There are other steps that could improve eDNA detection of amphibian chytrids. Although a detailed discussion is beyond the scope of this article, we want to mention other steps here, in order to promote further research. First, it is advisable to collect two filters per locality, as in the case of swabs [51]. Second, the TaqMan Environmental Master mix performs better than other options [49], without costing more than the approaches mentioned here (see Appendix A). Third, other PCR detection methods, such as digital-drop PCR, could be used as in [47].

We have confirmed *Bsal* in previous positive localities and we have announced four new positive sites in Spain. For the first time, an eDNA approach has been able to detect both pathogens responsible for chytridiomycosis. Our proposed eDNA collection procedure not only enables simultaneous sampling for *Bd* and for *Bsal*, but it is also at the moment the most economical option. We have also provided the number of ITS1 rRNA gene copies present in one cell, i.e., 120. This enables the genomic equivalent calculation to be refined. Our experiments showed that 96% ethanol was the best storage method. However, the use of silica gel provided comparable results, and we recommend this storage method because it is simpler and easier to use.

This research suggests a way to improve the work efficiency of field staff and reduces the amount of material required for field sampling, avoiding pumps or other electronic devices. For example, our method is easy to use in citizen science. In this way, eDNA analysis can act as an early warning system for invasive species. Our results contribute to optimize the resources in pathogen detection, and thus allow the early application of effective measures to mitigate the spread of chytridiomycosis disease.

## Figures and Tables

**Figure 1 jof-07-00258-f001:**
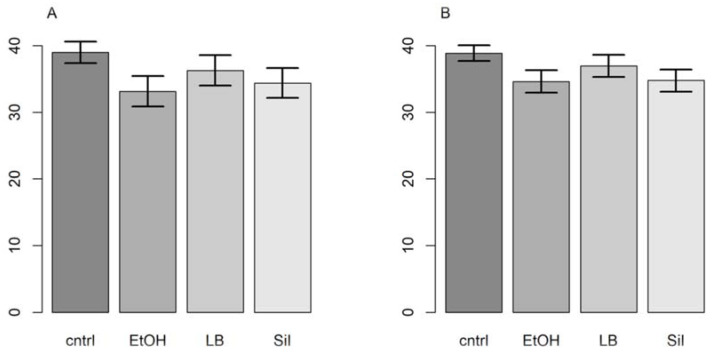
Ct values (Y axis) and storage methods (X axis). Control (cntrl), ethanol (EtOH), Longmire’s buffer (LB), and silica gel (Sil). The (**A**) plots (left side) are the primers from Blooi et al. (2013), and the (**B**) plots (right side) are from Spitzen-van der Sluijs et al. (2020).

**Table 1 jof-07-00258-t001:** List of *Batrachochytrium salamandrivorans* (*Bsal*) and *B.dendrobatidis* (*Bd*) positive localities in 2019. Number (N), (number of positive samples/number of collected samples). Distances between particular water bodies within the Teverga and Ruente localities are at least 5 km.

Country	Locality	First *Bsal* Positive	N Filters	N Swabs	*Bsal* DNA Copies	*Bd* (DNA Copies)
Spain	Ampuero	* 2017	1/1	5/5	10.81 ‡/8.05–23 §	
Spain	Teverga 1	* 2017	1/1	1/19	96.55 ‡/15.05 §	
Spain	Teverga 2	* 2017	1/1	0	15.70 ‡	
Spain	Teverga 3 †	2019	2/2	2/2	12.2–12.45 ‡/3.27–17 §	
Spain	Ruente 1	* 2017	1/2	0/20	12 ‡	
Spain	Ruente 2	2019	1/1	0/11	6.16 ‡	28.8 ‡
Spain	Suances	* 2017	2/2	0/25	22.35–96.05 ‡	31.2; 764.4 ‡
Spain	Ponga †	2019	2/2	0/15	1.51–24.2 ‡	
Spain	Cieza	2019	1/1	0/11	33.3 ‡	
Czech R.	Stare Mesto	N/A	2/2	11/30	N/A	0.1–7.3 ‡/1.91–44.55 §
Czech R.	Sokolov	N/A	0/32	N/A	N/A	

* *Bsal* positive localities from Lastra González et al., 2019. † Two different water habitats, but close to each other. ‡ are DNA copies—eDNA approach. § are DNA copies—swabs approach.

**Table 2 jof-07-00258-t002:** Our study comparison of the storage effect on eDNA capture (percentage of eDNA recovered from the original amount added) after six weeks of filters with different storage types (each type three filters) based on two quantification PCR assays from [19,21].

Storage Type	Primers from Blooi et al., 2013	Primers from Spitzen-van der Sluijs et al., 2020
Control, room temperature	0.01% *	0.1% *
Longmire’s buffer	0.04%	0.2% *
Silica gel	0.19%	1.1%
96% EtOH	0.34%	1.1%

* Not all wells amplified.

## Data Availability

Data is contained within the article or Appendix A.

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
