# Peer review of "Dual Detection of the Chytrid Fungi *Batrachochytrium* spp. with an Enhanced Environmental DNA Approach"

_jof, 2021, doi:10.3390/jof7040258_

Round 1

Reviewer 1 Report

In the manuscript “Dual detection of the chytrid fungi Batrachochytrium spp. With an enhanced environmental DNA approach”, the authors provide an enhanced (straightforward and cost effective) methodology for detection of both Bd and Bsal via eDNA water filtration collection and subsequent qPCR approach. They test and compare the method in the field and lab, including a comparison of filter storage methods, and an estimation of the number of ITS copies per Bsal cell using synthetic gBlocks.

Overall, the study is interesting, informative, with some novel elements, and the manuscript is well-written. My comments are mostly minor.

  • Lines 53-55: ‘Critical considerations stated in [23] should be followed…’. To be helpful for your readers, are you able to briefly overview these?
  • Line 62: Should this be ‘and the storage method’?
  • Line 67: ‘and multiple methods of sample preservation were tested’. Not sure which study you’re referring to here? Is this your study? Try to avoid passive voice.
  • Lines 85-88: ‘we have calculated a strict comparison of existing protocols’. Can you make it clearer here what you were actually comparing? Ie, costs, efficiency?
  • Lines 94-95: ‘We tested the filters in the Bsal-positive localities mentioned in [27]’. Can you clarify which these are? Can you also clarify the ‘four other localities’ in Spain?
  • Line 108: ‘Typical situation was…’. Sentence fragment.
  • Lines 112-113: Can you include the composition of Longmire’s buffer somewhere (perhaps in Suppl Info) or a clear reference for it. Can you also include a company name, location and product name for the silica gel (so other researchers can find a similar product). There are many products called ‘silica gel’, but also many different formulations (such as beads, etc).
  • Lines 143-144: Perhaps you could clarify your reason for testing with the [19] assay?
  • Line 145: ‘only if both wells amplified’. Perhaps I missed it earlier, but I didn’t see it clearly stated that the qPCR was performed in duplicate.
  • Lines 157-159: Can you clarify how you quantified the number of gBlock copies. With such small fragments I expect it would be difficult to obtain the level of precision you mention.
  • Lines 166-167: I’m guessing you mean that you produced 10-fold serial dilutions from starting concentrations ranging from 100-108 copies per uL? How many dilutions per starting concentration?
  • Line 171: Do you mean ‘In the case of performing the assay from [19]’?
  • Lines 180-182: Can you clarify what you mean by the ‘5*100…’ etc?
  • Line 198: Why the ‘x’ in ‘(LB x S x EtOH x control)’?
  • Line 201: ‘EtOH was 96% and control’ – sounds incomplete.
  • Line 258: Did you mean to include a comma between ‘40.221’ and ‘Rsquared’? Also, can you clarify the units for the ‘121.3’ value? Ie, is it copies per cell?
  • Line 265: It’s not clear to me what you mean here: ‘The overall gBlock capture…’
  • Line 273: Do you mean ‘with the primers from [19]’?
  • Lines 274-275: Can you provide effect sizes?
  • Line 284: ’22 euros per locality’. Do you mean ‘per sample collected’ (given that you later recommend collecting 2 filters per locality)?
  • Lines 297+: Can you start your Discussion section with a brief (perhaps 1 sentence) overview of the purpose of your entire study, then highlight your main results in the first paragraph before getting into discussing the details in later paragraphs? This should help readers who just skip to the Discussion section. Indeed, something along the lines of your paragraph from lines 424-432 would be helpful to start off the discussion section. Also, throughout your Discussion section, can you make it very clear which results are from your study, and which are from previous literature or other studies? At times it was a bit difficult to tell.
  • Line 300: Do you mean ‘it is becoming…’?
  • Line 314: ‘necessary to get four samples of 600 mL or five samples of 60 mL’. Can you clarify what you mean here and explain? Clearly these are different total volumes.
  • Line 367: Not sure what you mean by ‘based in [16]’?
  • Line 385: Did you mean to label this ‘Table 2’?

Author Response

Dear Reviewer,

Attached, you can find a point by point response to your comments.

Yours faithfully,

David

Reviewer 2 Report

Here is the review of the manuscript entitled " Dual detection of the chytrid fungi Batrachochytrium spp. with an enhanced environmental DNA approach" written by David Lastra González and co-authors.

The object of the paper was to test the use of eDNA for detecting aquatic chytrid fungi Batrachochytrium dendrobatidis (Bd) and B. salamandrivorans (Bsal) that are major threats to amphibians globally. The authors developed and tested a simple and cheap sampling protocol and the universally accepted qPCR assay aimed to detect the presence of both pathogenic species in previously identified sites in Spain as well as in few sites in Czech Republic. Their eDNA approach was efficient in the detection of both pathogenic species responsible for chytridiomycosis for the first time. Aditionally, they compared a few storage methods for samples and  the best method was to keep the samples in ethanol. But the use of silica gel provided similar results and they recommend this storage method because it is simpler and easier to use. The authors calculated the costs of their protocol and concluded that it is much cheaper than previously used ones. The results of this research could be used in monitoring of chytridiomycosis even by a citizen scientists. The paper makes very valuable and interesting contribution to the research field.

English language used is almost perfect. The overall study is conducted very well and I have only a few minor suggestions that are included in the pdf version of the paper. I recommend it for a publication in JoF.

Best wishes,

Reviewer

Author Response

Dear Reviewer,

Attached, you can find a response to your comments. To see the changes implemented check the new version of the manuscript. In summary, all your changes were accepted and modified according your suggestions.

Yours faithfully,

David
